# Investigating the Bactericidal Activity of an Ocular Solution Containing EDTA, Tris, and Polysorbate 80 and Its Impact on the In Vitro Efficacy of Neomycin Sulfate against *Staphylococcus aureus*: A Preliminary Study

**DOI:** 10.3390/antibiotics13070611

**Published:** 2024-06-30

**Authors:** Sophie Amiriantz, Sara Hoummady, Elodie Jarousse, Séverine Roudeix, Thomas Philippon

**Affiliations:** 1Dômes Pharma, ZA Champ Lamet, 3 Rue André Citroën, 63430 Pont-du-Château, France; 2Transformations et Agro-Ressources, ULR 7519, Institut Polytechnique Unilasalle—Collège Vétérinaire, Université d’Artois, 76130 Mont Saint Aignan, France; 3Groupe Icare, Biopôle Limagne, 6 Rue Emile Duclaux, 63360 Saint Beauzire, France

**Keywords:** *Staphylococcus aureus*, neomycin sulfate, ocular surface infection, antimicrobial resistance, treatment

## Abstract

In the current context of emerging and spreading antimicrobial resistance in human and animal infections, new strategies need to be developed to improve the efficacy of commonly prescribed antibiotics and preserve more critical compounds for multi-drug-resistant infections. This preliminary study aimed at evaluating the benefits of an eye cleaning solution containing 0.1% EDTA, 0.02% Tris, and 0.1% Polysorbate 80 in veterinary ophthalmology. A first in vitro study was performed to assess the bactericidal activity of the test solution against *Staphylococcus aureus* and *Pseudomonas aeruginosa* strains. A second in vitro study evaluated the impact of the test solution on the antimicrobial activity of neomycin against *Staphylococcus aureus*. The test solution alone did not show bactericidal activity against *Staphylococcus aureus* and *Pseudomonas aeruginosa*. The test solution seemed to increase the activity of Neomycin Sulfate against *Staphylococcus aureus*. These findings warrant further research to better characterize the impact on the bactericidal activity of antimicrobials used in veterinary ocular surface infections of the solution containing 0.1% EDTA, 0.02% Tris, and 0.1% Polysorbate 80 as well as of each individual ingredient for a thorough understanding of how this test solution could provide a new strategy to address the growing antimicrobial resistance issue worldwide.

## 1. Introduction

Infectious keratitis in human patients is a severe, vision-threatening ophthalmic emergency described worldwide. The two bacterial species most commonly identified are Gram-positive *Staphylococcus aureus* and Gram-negative *Pseudomonas aeruginosa*, both of which are increasingly resistant to antimicrobials [1]. Given the increasing burden of antimicrobial resistance and the slow rate at which new antimicrobial classes are developed and reach the market [2,3], the need for new therapeutic options to treat bacterial keratitis is becoming more pressing.

Ocular surface infections are also frequent in veterinary ophthalmology, and can lead to severe complications threatening the patients’ vision and ocular integrity [4,5,6,7,8,9]. The most common bacterial species identified in canine and equine ocular infections are *Staphylococcus pseudintermedius* [7,10,11,12,13,14] and other strains of *Staphylococcus* spp. [10,11,12,13,14,15], *Streptococcus* spp. [11,13,14,15], *Enterobacter* spp. [12,15], and *Pseudomonas aeruginosa* [10,11,12,13,14,15]. In cats, *Staphylococcus* spp. [9,14], *Chlamydia felis*, *Mycoplasma* spp., *Bordetella bronchiseptica* [16], *Streptococcus* spp., and *Pasteurella* spp. [14] are the most frequent pathogens identified from bacterial culture of corneal or conjunctival samples. Some of these bacterial species are associated with keratomalacia (or melting ulcer) [17], a vision-threatening condition where the corneal proteins are degraded faster than they are renewed, which requires urgent and aggressive medical or surgical treatment [18,19].

The two most prevalent bacterial species encountered in canine, feline, and equine ocular surface infections, namely *Staphylococcus* spp. and *Streptococcus* spp., have a single-layered membrane [20]. The other bacteria most commonly cultured from ocular surface disease samples (i.e., *Enterobacter* spp., *Pseudomonas* spp., *Chlamydia* spp., *Bordetella bronchiseptica*, *Pasteurella multocida*) belong to the Gram-negative group [20]. The outer wall of Gram-negative bacteria is thought to result from an evolutionary process undergone by bacteria with what was initially a single cell membrane. This evolutionary step is suspected to stem from the antibiotic selection pressure to which non-antibiotic-producing bacteria were submitted, enabling their survival [21]. This would explain why the antimicrobial susceptibility profile of Gram-negative bacteria is usually narrower than that of Gram-positive bacteria.

Antibiotic resistance exerted by both Gram-positive and Gram-negative bacteria is an emerging concern in veterinary ophthalmology, as shown by recent epidemiological studies detailing the evolution of antimicrobial susceptibility profiles of pathogens cultured from canine, feline, and equine patients worldwide [10,11,13,14,15,22]. This is all the more concerning in that antibiotic resistance of ocular bacteria was shown to be associated with ocular disease [12].

The same study also demonstrated that the biofilm-forming ability of bacteria was associated with ocular disease [12]. When organized into a biofilm, bacteria are less easily reached by antibiotics and antimicrobial tear film compounds, due to the extracellular polymeric substance (EPS) they secrete and are embedded in [20]. Along with innate and acquired antibiotic resistance factors, biofilms reduce the efficacy of antibiotics and pose a serious health risk in veterinary and human medicine.

In order to limit the emergence of resistance to the critical antimicrobials that are the last option to treat infections involving multi-drug-resistant bacterial pathogens in humans, the Committee for Medicinal Products for Veterinary Use (CVMP) and the Committee for Medicinal Products for Human Use (CHMP) established an Antimicrobial Advice Ad Hoc Expert Group (AMEG) to issue a Categorisation of antibiotics in the European Union [23]. This report is destined to serve as a guideline for a reasonable use of antimicrobials in veterinary and human medicine. Antimicrobial compounds are thus classified into four categories, depending on how critical they are for multi-drug resistant (MDR) infections in humans: category D, “use with prudence”; category C, “use with caution”; category B, “restrict”; and category A, “avoid”. This classification encourages veterinarians to rationalize their prescription of antibiotics, including when administered topically in the eye and to individual companion animals, and for antimicrobial compounds usually used empirically as first-line treatment or to prevent an infectious complication of ulcerative keratitis. Based on this classification, most antimicrobials available in veterinary ophthalmology in different geographies are to be used “with prudence” (cat. D: fusidic acid, tetracyclines, bacitracin) or “with caution” (cat. C: chloramphenicol, aminoglycosides (neomycin, framycetin, gentamicin, tobramycin), with the exception of polymyxin B, the prescription of which should be “restricted” (cat. B).

When presented to second- or third-opinion practices, a majority of patients with ocular surface infections have already been prescribed at least one topical antibiotic that did not lead to clinical improvement [8,17]. This seems to highlight the need to improve the microbial response to the existing topical ocular antibiotics available in veterinary medicine, to avoid having to resort to higher concentrations of antimicrobials, or to antimicrobials usually classified in categories C (“use with caution”: first- and second-generation cephalosporins such as cefazolin; aminoglycosides such as tobramycin; macrolides such as erythromycin) or B (“restrict”: third- and fourth-generation cephalosporins such as ceftiofur; quinolones such as enrofloxacin and moxifloxacin).

In view of the limited innovation in the area of ocular antimicrobial therapy in veterinary medicine, leading veterinarians to resort to increased frequencies of administration, increased eye drop concentrations, or molecules not licensed in veterinary ophthalmology, solutions to improve the efficacy of existing options are required. This would help limit the escalation in antimicrobial use in veterinary ophthalmology, and potentially prevent severe complications of ocular surface infections, leading to loss of vision or even enucleation in veterinary patients.

Ethylenediaminetetraacetic acid (EDTA) is a mineral- and metal-chelating agent used in numerous ocular preparations and in other medical areas where biofilms can occur. Trisaminomethane (Tris) is a compound used as an alkaline buffer in many solutions that potentiates the chelating action and antibiofilm activity of EDTA [24]. The use of EDTA and Tris, combined with topical antibiotics, has been shown in vitro to increase the efficacy of antimicrobials against selected bacterial species responsible for otitis externa in companion animals [25,26,27,28,29,30,31,32]. Additionally, the combination of the two compounds has shown more activity than each compound taken separately [24]. These results have led to the commercialization of several ear cleaners containing Tris-EDTA. Tris, EDTA, and Tris-EDTA have also been shown to reduce the biofilm-forming abilities of *Pseudomonas aeruginosa* and *Staphylococcus* spp., and to enhance antibiotic activity on the growth and survival of those biofilms [24,30,33]. Moreover, synergy between Tris-EDTA and Amikacin or Neomycin Sulfate was evidenced in vitro by Sparks and colleagues [27].

Polysorbate 80 (P80) is a surfactant emulsifier, commonly used in cosmetics and food to prevent bacterial contamination or facilitate mixing lipophilic and aqueous substances [34]. Polysorbate 80 has also demonstrated anti-biofilm activities [35,36,37].

A veterinary eye cleaning solution containing EDTA, Tris, and P80 is available in Europe, to clean the eyes of veterinary patients and prepare the ocular surface for subsequent topical treatments. In this preliminary, non-controlled in vitro study, the authors aimed to investigate the bactericidal activity of the ingredients of this commercial eye cleaning solution combined, and to establish the proof of concept of evaluating the potential synergy between the test solution and Neomycin Sulfate, an AMEG C antibiotic contained in several ophthalmic drugs, on bacterial growth and survival. We hypothesize that the commercial eye cleaning solution containing Tris, EDTA, and Polysorbate 80 has no bactericidal activity on *Pseudomonas aeruginosa* and *Staphylococcus aureus*, but might increase the in vitro bactericidal activity of Neomycin Sulfate against *Staphylococcus aureus*.

## 2. Results

### 2.1. Effect of the Test Solution on Bacterial Growth

Increasing concentrations of the ophthalmic solution containing 0.1% EDTA, 0.02% Tris, and 0.1% Polysorbate 80 failed to achieve a more-than-5-log reduction in the number of viable colonies of *Staphylococcus aureus* CIP 4.83 (ATCC 6538) and *Pseudomonas aeruginosa* CIP 103467 (ATCC 15442) when added to the culture medium as shown in Table 1 and Table 2, respectively.

This confirms our hypothesis that the aqueous solution containing 0.1% EDTA, 0.02% Tris, and 0.1% P80 does not present bactericidal activity on its own.

#### 2.1.1. *Staphylococcus aureus* ATCC 6538

The number of viable microorganisms was reduced by less than 4 log in the presence of 10%, 50%, or 80% of the test solution compared to the control, as shown in Table 1.

#### 2.1.2. *Pseudomonas aeruginosa* ATCC 15442

The number of viable microorganisms was not reduced by more than 5 log in the presence of 10%, 50%, or 80% of the test solution compared to the control, as shown in Table 2.

### 2.2. Effect of the Test Solution in Combination with Neomycin Sulfate on Bacterial Growth

The bactericidal activity of Neomycin Sulfate against *Staphylococcus aureus* CIP 53.156 was increased when the Neomycin Sulfate eye drops were mixed with the test solution in a 1:1 ratio, as shown in Table 3 and Figure 1 below.

## 3. Discussion

This preliminary, non-controlled study aimed at investigating the in vitro effect of an eye cleaning solution containing 0.1% EDTA, 0.02% Tris, and 0.1% Polysorbate 80 against three bacterial isolates in the absence and in the presence of Neomycin Sulfate. Two main results emerged from this study. In the first in vitro study, the test solution was shown to exert no bactericidal action on *Staphylococcus aureus* and *Pseudomonas aeruginosa*. In the second, proof-of-concept in vitro study, the test solution seemed to potentiate the activity of Neomycin Sulfate against *Staphylococcus aureus*.

The first in vitro study showed that a 0.1% EDTA, 0.02% Tris, and 0.1% Polysorbate 80 solution did not reduce the bacterial counts of *Staphylococcus aureus* and *Pseudomonas aeruginosa* by more than 5 log, which is the threshold defined by the regulatory bodies to categorize a substance as bactericidal. The design of the experiment did not allow for a finer assessment of the bacterial counts in the presence of 10%, 50%, and 80% concentrations of the test solution. Additional dilutions of the suspensions would have been required for the numeration of CFU; however, the initial aim of the experiment was to assess the positioning of the test solution with regards to the regulation on biocides. Banin and colleagues [24] showed that EDTA at 50 mM (equivalent to 1.46% *m*/*v*) kills planktonic *Pseudomonas aeruginosa* as well as affects *Pseudomonas aeruginosa* biofilm-forming abilities. Additionally, they showed that combining EDTA at 50 mM with Tris at 20 mM (equivalent to 0.24% *m*/*v*) further increased the bactericidal and antibiofilm activity of EDTA at 50 mM against *Pseudomonas aeruginosa*. Later, Toutain-Kidd and colleagues [35] studied the effect of Polysorbate 80 concentrations ranging from 0.001% to 0.1% on *Pseudomonas aeruginosa* growth and biofilm formation. Their results showed that concentrations as low as 0.001% decreased biofilm formation, but that even the highest P80 concentration (0.1%) did not inhibit planktonic growth of *Pseudomonas aeruginosa*. The solution tested in the present study contained concentrations of EDTA and Tris 14.6 and 12 times lower than those tested in the study from Banin et al. [24], respectively. This may explain why the tested solution did not reduce the planktonic bacterial counts of either *Staphylococcus aureus* or *Pseudomonas aeruginosa*, since Polysorbate 80 is not expected to present bactericidal activity.

The second in vitro study measured the activity of Neomycin Sulfate in the absence and in the presence of the test solution at a 1:1 ratio. The activities of Neomycin Sulfate in the absence and in the presence of the test solution are 3267 IU/mL and 3708 IU/mL, respectively. The increase in antimicrobial activity provided by the test solution is approximately 13.5%, indicating that the test solution might potentiate the activity of Neomycin Sulfate against Gram-positive *Staphylococcus aureus*. These findings require confirmation with a larger-scale, controlled study involving a higher number of replicates, to investigate the repeatability and reliability of the present results. Furthermore, the contribution of each individual ingredient to the results needs to be assessed. Previously mentioned studies [24,35] only tested the compounds on Gram-negative *Pseudomonas aeruginosa*, and showed that Tris and EDTA enhanced the bactericidal and antibiofilm action of gentamicin against *Pseudomonas aeruginosa*. One recent in vitro study assessed the impact of Tris-EDTA (0.225–0.06%) on the antimicrobial activity of topical otological treatments against multi-drug-resistant *Staphylococcus pseudintermedius* isolates [29]. The results indicated that the action of Tris and EDTA was not limited to Gram-negative bacteria, as it reduced the minimal inhibitory concentration (MIC) of gentamicin against *Staphylococcus aureus*. However, the MIC of the other antimicrobials (to most of which the isolates were resistant) was not altered by the presence of Tris and EDTA. In addition, the minimal bactericidal concentration (MBC) of gentamicin was not modified in the presence of Tris and EDTA. Conversely, the present study highlighted that the test solution potentiated the in vitro bactericidal activity of Neomycin Sulfate against a *Staphylococcus aureus* isolate. In the test solution, the concentrations of Tris and EDTA were, respectively, 11.25 times higher and 1.7 times lower than in the test solution of the present study. Additionally, the antimicrobial susceptibility profile of the isolate used was not known. The results might have been different with a bacterial isolate known to be resistant to Neomycin Sulfate.

The design of this study, a turbidimetric method to measure the potency of an antibiotic substance compliant with the European Pharmacopoeia for the microbiological assay of antibiotics, might not have been the most adequate for research purposes, as it did not allow for a control group, nor for serial testing. This test does not provide the possibility to measure the bactericidal activity of a negative control, as the control replicate without the antibiotic serves as a reference to compare the optical densities of the replicates containing the antibacterial substance, from which the potency of the antibiotic is calculated [38]. In addition, only one antibiotic activity value was obtained for each of the study conditions (Neomycin Sulfate alone, and Neomycin Sulfate combined with the test solution in a 1:1 ratio), which did not allow for a statistical comparison of the results. Consequently, the 13.5% difference observed between Neomycin Sulfate and the combination of Neomycin Sulfate with the test solution in a 1:1 ratio might simply represent the intra-assay variability of the method rather than an actual difference in antibiotic potencies. A different assay, such as diffusion, the disc method, or bacterial counts in all replicates, would have allowed for proper control and a larger sample size, and a subsequent statistical analysis.

The exact mechanism by which the test solution reduced the concentration of Neomycin Sulfate required to inhibit the growth of *Staphylococcus aureus* has not been investigated by the authors. However, evidence in the scientific literature can help formulate an explanation. The concentrations of Mg^2+^ and Ca^2+^ in the culture medium of *Pseudomonas aeruginosa* affect the susceptibility of the bacteria to gentamicin, due to those divalent cations inhibiting the binding of gentamicin to the bacterial surface [39]. This inhibition process is the consequence of a competition of Mg^2+^ and Ca^2+^ with aminoglycosides for binding sites located at the surface of the cell wall or cytoplasmic membrane of Gram-positive and Gram-negative bacteria [28,40]. Additionally, Mg^2+^ and Ca^2+^ antagonize the bacterial uptake of aminoglycosides [40]. In 1984, Wooley and colleagues [26] measured the antibiotic uptake of Gram-negative bacteria after exposure to a solution containing 3.22 mM EDTA and 50 mM Tris. The results indicated that pre-treating *Escherichia coli*, *Proteus vulgaris*, and *Pseudomonas aeruginosa* with Tris-EDTA increased the intracellular concentrations of the antibiotics to which the bacteria were then exposed. The authors suggest two mechanisms by which the results can be explained. In *Escherichia coli* and *Pseudomonas aeruginosa*, Tris-EDTA was considered to have increased bacterial permeability to antibiotics. In *Proteus vulgaris*, Tris-EDTA was thought to have inhibited the efflux mechanism of the bacteria, responsible for decreasing intracellular concentrations of antibiotics. In our experiment, in addition to changing cell wall permeability [28], EDTA and Tris could have decreased the concentrations of Mg^2+^ and Ca^2+^ immediately surrounding *Staphylococcus aureus* through their chelating action, thus facilitating the binding of Neomycin Sulfate to the bacterial surface, and its subsequent uptake. To better understand how the eye cleaning solution altered the in vitro efficacy of Neomycin Sulfate against *Staphylococcus aureus*, further studies evaluating the binding of Neomycin Sulfate to the bacterial surface and the bacterial concentrations of Neomycin Sulfate in the absence and in the presence of the test solution should be conducted.

The bacterial strains used in these studies, *Staphylococcus aureus* ATCC 6538, *Staphylococcus aureus* CIP 53.156, and *Pseudomonas aeruginosa* ATCC 15442, were purchased from a bacterial strain provider. These may have poorly represented the bactericidal activity of the eye cleaning solution, or the activity the antibiotics might have had against a bacterial isolate obtained from a veterinary patient with ocular surface infection. However, one of the objectives of this study was to compare the activity of the antibiotic alone and with the test solution. It would have been preferrable to conduct this study with bacterial isolates obtained in a clinical setting, but in our opinion, the bias does not jeopardize the conclusions of this study. Moreover, the benefit of using standardized strains lies in the repeatability of tests and a better comparison of the effect of potential future products.

Whether the bacterial strains used in these studies had biofilm-forming abilities is unknown. Therefore, the theoretical ability of Tris, EDTA, and Polysorbate 80 to prevent biofilm growth and survival could not be tested. Future studies with the test product should be conducted on bacterial strains with biofilm-forming abilities, for a better understanding of the extent of action of the combination of Tris, EDTA, and Polysorbate 80 at concentrations of 0.02, 0.1, and 0.1%, respectively.

The present studies were performed in vitro, in conditions that do not mimic ocular infections nor the dynamics of tear film production, evacuation, and renewal. The tear film contains antimicrobial substances, such as albumin, lysozymes, lactoferrin, or immunoglobulins [41]. As shown by Sebbag and colleagues [42], the protein content of the tear film impacts the fraction of an unbound antibiotic that is microbiologically active, and could lead to drug inefficacy despite a correct amount of antibiotic administered onto the eye. It would therefore have been interesting to add, e.g., albumin to the culture medium, and assess if the adjunction of the test solution would have counteracted the binding of Neomycin Sulfate to albumin and still increased its activity. This in vitro setting could not evaluate if the sequential instillation of the test solution and the antibiotic approximately 5 min apart, as is usually recommended when several topical treatments have to be administered [43], would have altered or enhanced the synergy between the test solution and Neomycin Sulfate eye drops. In particular, one could expect the instillation of the test solution to force the evacuation of the tear film, and therefore reduce the protein content of the film covering the ocular surface prior to antibiotic administration. It is also unknown if the action of the test solution would have persisted until the administration of the antibiotic. The synergy observed in vitro between the eye cleaning solution and the Neomycin Sulfate eye drops could be enhanced or reduced in an in vivo setting, which remains to be elucidated.

Commercial eye drops containing Neomycin Sulfate and polymyxin B (Tévémyxine^®^ collyre/Duomyxin^®^, Dômes Pharma, Pont-du-Château, France) were chosen to test the synergy between the eye cleaning solution and an antibiotic traditionally used against *Staphylococcus* spp. in canine, feline, and equine ocular infections. Neomycin Sulfate is indeed present in several ophthalmic preparations, and always combined with polymyxin B, polymyxin B and bacitracin (more frequently referred to as the “triple antibiotic”), or other antibiotics and steroids. Testing the Neomycin Sulfate + polymyxin B combination in our study rather than Neomycin Sulfate alone is not expected to have impacted the results as polymyxin B targets Gram-negative bacteria (e.g., *Pseudomonas aeruginosa*) and is not active against *Staphylococcus aureus*. However, it would have been interesting to assess whether the synergy between Neomycin Sulfate and bacitracin that has been described [44] would have been further enhanced by the eye cleaning solution.

The Neomycin Sulfate/test solution ratio in our study was 1:1. It would be relevant to measure the actual ratio of the antibiotic eye drop/tear film when an antibiotic eye drop is administered onto the eye of a patient. This would enable us to establish the optimal test solution/antibiotic ratio that would provide the best synergistic effect while ensuring that the quantity of antibiotic administered allows for bactericidal/bacteriostatic concentrations in the tear film.

The two studies described in this article provide preliminary data regarding the benefit of using an eye cleaning solution containing 0.02% Tris, 0.1% EDTA, and 0.1% Polysorbate 80 prior to topical antibiotic administration in ocular surface infections. These studies were performed with bacterial strains that were not obtained on veterinary patients with ocular infections. Future similar studies should use more clinically relevant bacterial isolates, such as *Pseudomonas aeruginosa*, *Staphylococcus pseudintermedius*, *Streptococcus* spp., *Mycoplasma* spp., and *Chlamydia felis*, obtained in canine and feline patients with ocular infection.

Similarly, the activity of other antibiotics commonly prescribed in veterinary ophthalmology and pertaining to the D (“use with prudence”) or C (“use with caution”) AMEG categories (fusidic acid, tetracyclines, chloramphenicol, other aminosides, first- and second-generation cephalosporins) should be assessed in the absence and in the presence of the test solution. This would provide a more comprehensive panel of the benefits provided by the Tris, EDTA, and Polysorbate 80 eye cleaning solution.

Mostly, the applicability of the results obtained in in vitro to in vivo situations remains to be determined. The feasibility of such comparisons and assessments in veterinary patients with spontaneous ocular infection could be challenging, due to the heterogeneity of ocular infections, individual responses to medical treatment, and overall owner compliance [45,46,47].

Other non-antimicrobial strategies have been described to treat infectious ocular surface diseases. A recent study by Walter and colleagues [48] has shown that N-acetylcysteine had antimicrobial activity against clinical isolates of *Staphylococcus pseudintermedius*, *Streptococcus canis*, and *Pseudomonas aeruginosa* at concentrations as low as 0.156%. Corneal cross-linking (“photoactivated chromophore for keratitis–corneal crosslinking”, PACK-CXL) has also been demonstrated to exert bactericidal activity [49,50]. This method requires UV-A light and riboflavin, and its bactericidal activity stems from riboflavin inserting in the bacterial genome, ultimately damaging it. PACK-CXL overcomes antimicrobial resistances due to its different targets and mechanism of action. However, it requires specific equipment that can prove to be expensive and less accessible to first-opinion practices. Another method uses UV-C light instead of UV-A light, and has shown in vitro, ex vivo, and in vivo efficacy against bacterial pathogens [51].

In the current context of the emergence and spreading of antimicrobial resistance against commonly and less commonly used antibiotic compounds, new strategies to replace or reinforce antibiotics in veterinary ophthalmology are required. Treatment adjuvants such as an ocular cleaning solution containing Tris, EDTA, and Polysorbate 80 could prove to be a valuable asset in veterinary ophthalmology and mitigate the complication risk leading to vision loss or enucleation. Some bacterial species (*Pseudomonas aeruginosa*, β-hemolytic *Streptococcus* spp.) have been identified as a risk factor for keratomalacia, a phenomenon leading to rapid worsening of corneal ulcers and the possible destruction of the ocular globe [17], which requires aggressive medical treatment, or surgery [18,19]. Cleansing the ocular surface with a solution containing 0.02% Tris, 0.1% EDTA, and 0.1% Polysorbate 80 could increase the permeability of bacteria involved in ulcerative disease to antibiotics administered topically. Reducing the concentrations of antibiotics required on the ocular surface to achieve bacterial eradication could contribute to preventing corneal ulcers from evolving into melting ulcers, and thus limit vision loss or the need for enucleation in veterinary patients.

## 4. Materials and Methods

### 4.1. Test Product

The initial test product is an ophthalmic aqueous solution containing 0.1% ethylenediaminetetraacetic acid (EDTA), 0.02% Trisaminomethane (Tris), and 0.1% Polysorbate 80 (P80) (OphtaPRIME^®^, Dômes Pharma, Pont-du-Château, France). Its pH ranges from 7.2 to 7.6, and its osmolality is between 230 and 300 mOsm/kg (Dômes Pharma internal data).

### 4.2. Neomycin Sulfate Eye Drops

The antibiotic eye drops used in our study were a commercial combination of Neomycin and Polymyxin B (Tévémyxine^®^ collyre/Duomyxin^®^, Dômes Pharma, Pont-du-Château, France) presenting as a lyophilisate (Neomycin at 17,000 IU and Polymyxin B at 50,000 IU) to be reconstituted in 5 mL of a solvent. The final solution contains 3400 IU of Neomycin and 10,000 IU of Polymyxin B per milliliter. It is licensed in the European market to treat ocular infections caused by bacteria sensitive to Polymyxin and Neomycin in cats and dogs.

### 4.3. Bacterial Strains

This study used the following commercial strains: *Staphylococcus aureus* ATCC 6538, *Staphylococcus aureus* CIP 53.156, and *Pseudomonas aeruginosa* ATCC 15442 (Collection de l’Institut Pasteur, Paris, France).

### 4.4. Evaluation of the Bactericidal Activity of the Test Product

The evaluation of the bactericidal activity of the test product complied with the NF EN ISO 1040 standard [52].

#### 4.4.1. Preparation of the Test Product Concentrations

The ready-to-use test product was diluted extemporaneously in sterile purified water to reach concentrations of 12.5%, 62.5%, and 100% (corresponding to 10%, 50%, and 80% during testing), and used within 2 h for testing.

#### 4.4.2. Preparation of the Bacterial Suspensions

*Pseudomonas aeruginosa* ATCC 15442 and *Staphylococcus aureus* ATCC 6538 were grown on trypase soy agar for 18 to 24 h at 36 °C ± 1 °C, then subcultured once or twice for 18 to 24 h at 36 °C ± 1 °C. The working subculture was suspended in 10 mL of a tryptone salt solution in a tube containing 5 g of sterile glass beads and homogenized by vortexing. Optical density was used to adjust the titer of the suspension between 1.5 × 10^8^ CFU/mL and 5.0 × 10^8^ CFU/mL. The trial suspension was kept at 20 °C ± 1 °C and used within 2 h of preparation.

#### 4.4.3. Conduct of the Bactericidal Activity Trial

Eight milliliters of each concentration of the test product were added to 1 mL of the bacterial suspension mixed with 1 mL of sterile purified water, to reach final concentrations of 10%, 50%, and 80% of the test product. The test product was maintained in contact with the bacterial suspensions at 20 °C ± 1 °C for 5 min ± 10 s.

One milliliter of the test product + bacterial suspension was then added to 8 mL of a neutralizer and 1 mL of sterile purified water, and the new mix was maintained at 20 °C ± 1 °C for 5 min ± 10 s. The neutralized mix was then deposited by the enumeration pour plate method and incubated for an additional 40–48 h prior to final CFU counting. For each concentration and each bacterial suspension, the experiment was performed in duplicate.

No positive or negative control was performed in this study. The number of viable microorganisms for each concentration of the test product was compared to the number of viable microorganisms in the initial bacterial suspension.

### 4.5. Evaluation of the Antimicrobial Activity of the Eye Drops and Test Product

The evaluation of the antimicrobial activity of the antibiotic eye drops alone and mixed with the test product (1:1 ratio) was performed by microbiological assays of antibiotics carried out with the turbidimetric method following the European Pharmacopoeia 10th edition, chapter 2.7.2 [38]. For each testing condition (antibiotic eye drops alone or mixed with the test product), 8 replicates were performed.

#### 4.5.1. Preparation of the Eye Drop Dilutions

The eye drop solution to be tested was diluted in purified water to obtain a concentration of approximately 25 IU/mL, further diluted down to 15 IU/mL, 9 IU/mL, and 5 IU/mL.

#### 4.5.2. Preparation of the Bacterial Suspensions

*Staphylococcus aureus* CIP 53.156 was grown on agar medium 1 for 18 to 24 h at 36 °C ± 1 °C. After incubation, a suspension was prepared in medium C with an optical density of approximately 0.250 measured at a wavelength of 620 nm. The exact titer of the suspension was determined by the enumeration pour plate method in agar medium 1.

#### 4.5.3. Conduct of the Eye Drop Antimicrobial Activity Trial

One milliliter of the eye drop solution was then added to 9 mL of contaminated medium C, resulting in final antibiotic concentrations of 2.5 IU/mL, 1.5 IU/mL, 0.9 IU/mL, and 0.5 IU/mL. The tubes containing the antibiotic and contaminated medium C were incubated at 36 °C ± 1 °C for 4 h; then, microbial growth was blocked by adding formaldehyde to each tube in a cold-water bath (2–8 °C) and left in contact for a minimum of 10 min.

The inhibition of microbial growth by the antibiotic was determined by measuring the optical density of each tube at a wavelength of 620 nm.

The activity of the antibiotic was calculated using the following formula:(1)$$Activity\;A\left(\frac{IU}{mL}\right)=\frac{Activity\;(\%)}{100}\times \frac{TS\;(IU)}{Vf\;(mL)}\times \frac{VS\;(mL)}{Vf\;(mL)}\times \frac{Vf\;(mL)}{PU\;(mL)}\times \frac{Vf\;(mL)}{VU\;(mL)},$$
with

Vf: volume of the vial.

TS: titer of the standard.

VS: volume of stock solution to obtain a 25 IU/mL dilution of the standard.

PU: test sample of the product to be tested.

VU: volume of stock solution taken to obtain a 25 IU/mL dilution of the eye drops.

#### 4.5.4. Conduct of the Potentiation of the Antimicrobial Activity of Neomycin Sulfate Eye Drops by the Test Solution Trial

This trial was conducted with the same protocol as for the eye drop antimicrobial activity trial. For each concentration of the eye drop solution, the experiment was performed in 8 replicates.

No negative or positive control was performed in this study. The activity of the antimicrobial alone was compared to the activity of the antimicrobial mixed with the test solution at a 1:1 ratio.

## 5. Conclusions

An ophthalmic aqueous solution containing 0.1% EDTA, 0.02% Tris, and 0.1% Polysorbate 80 does not present bactericidal nor bacteriostatic activity in vitro against *Pseudomonas aeruginosa* and *Staphylococcus aureus*.

The same aqueous solution seems to increase the in vitro activity of Neomycin Sulfate against *Staphylococcus aureus* in a 1:1 ratio combination.

These findings warrant further research to better characterize the impact on the bactericidal activity of antimicrobials used in veterinary ocular surface infections of the solution containing 0.1% EDTA, 0.02% Tris, and 0.1% Polysorbate 80 as well as of each individual ingredient for a thorough understanding of how this test solution could provide a new strategy to address the growing antimicrobial resistance issue worldwide.

## Figures and Tables

**Figure 1 antibiotics-13-00611-f001:**
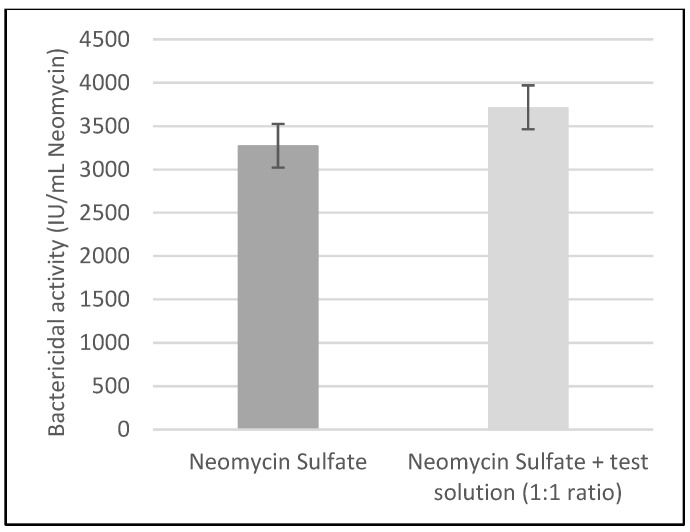
Activity of Neomycin Sulfate and Neomycin Sulfate + the test solution (1:1 ratio) measured by the turbidimetric method in a suspension of *Staphylococcus aureus* CIP 53.156. The quantity of Neomycin Sulfate added to the bacterial suspension in the presence of the test solution at a 1:1 ratio is half that in the absence of the test solution. The vertical bars represent the 95% confidence intervals.

**Table 1 antibiotics-13-00611-t001:** The amount of viable *Staphylococcus aureus* ATCC 6538 in the presence of 10%, 50%, or 80% of the test solution compared to the control.

	Tested Concentrations			
	Initial Bacterial Suspension	10%	50%	80%
Number of viable microorganisms (CFU ^1^/mL)	2.56 × 10^7^	>3.3 × 10^3^	>3.3 × 10^3^	>3.3 × 10^3^
Logarithmic reduction in the number of viable microorganisms at the trial solution concentration	N/A	<3.9	<3.9	<3.9

^1^ CFU: Colony-Forming Unit.

**Table 2 antibiotics-13-00611-t002:** The amount of viable *Pseudomonas aeruginosa* ATCC 15442 in the presence of 10%, 50%, or 80% of the test solution compared to the control.

	Tested Concentrations			
	Initial Bacterial Suspension	10%	50%	80%
Number of viable microorganisms (CFU/mL)	2.92 × 10^7^	>3.3 × 10^3^	>3.3 × 10^3^	>3.3 × 10^3^
Logarithmic reduction in the number of viable microorganisms at the trial solution concentration	N/A	<4.0	<4.0	<4.0

**Table 3 antibiotics-13-00611-t003:** Turbidimetric assay results with *Staphylococcus aureus* CIP 53.156 in the presence of Neomycin Sulfate eye drops alone or mixed with the test solution at a 1:1 ratio.

	Activity Obtained (IU ^1^/mL Neomycin Sulfate)	95% Confidence Interval (IU/mL Neomycin Sulfate)
Neomycin Sulfate eye drops	3267	3024–3527
Neomycin Sulfate eye drops + test solution (1:1 ratio)	3708	3554–3872

^1^ IU: International Unit.

## Data Availability

The original contributions presented in the study are included in the article; further inquiries can be directed to the corresponding authors.

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
