# Peer review of "Investigating the Bactericidal Activity of an Ocular Solution Containing EDTA, Tris, and Polysorbate 80 and Its Impact on the In Vitro Efficacy of Neomycin Sulfate against Staphylococcus aureus: A Preliminary Study"

_antibiotics, 2024, doi:10.3390/antibiotics13070611_

Round 1
Reviewer 1 Report
Comments and Suggestions for Authors
I have reviewed the manuscript titled "An ocular solution containing EDTA, Tris, and Polysorbate 80 increases the in vitro efficacy of neomycin sulfate against Staphylococcus aureus but does not exert bactericidal activity on its own" The authors examined the effects of 0.1% EDTA, 0.02% Tris, and 0.1% Polysorbate 80 in an eye cleaning solution for assessing the bactericidal activity. The paper is acceptable however it needs some corrections/Modifications before publication.
1. Title: The title of your manuscript should be concise, specific and relevant. The title should also convey the purpose and/or key findings of the research.
2. Abstract: It's recommended that authors update their abstracts to ensure they include all important content. The abstract serves as an objective summary of the article, so it's important to include key information.
3. Introduction: In lines 122-125, there's a hypothetical sentence of a potential synergistic effect of neomycin sulfate with solutions containing Tris, EDTA, and Polysorbate 80. However, these data haven't been previously mentioned. To ensure completeness of the content, the author should consider adding relevant information on this topic.
4. Authors are encouraged to include details about the positive and negative controls utilized in the experiment for clarity and completeness.
5. Result: The author is encouraged to enhance the reliability of the experimental results by employing statistical analysis or calculations to interpret the data more effectively.
6. Result: No subtopics 2.1.1
7. Method: It would be beneficial for authors to prepare their own test solutions rather than solely relying on commercially available products. Additionally, increasing the number of samples used in the experiment would enhance the robustness of the study.
8. References: It's recommended to update some of the older references with more recent ones, preferably within the last 10 years, to ensure the inclusion of the latest research findings.
Some minor errors were detected in the text. Minor editing of English grammar required.
Author Response
Please see the attachement the response to your comments and suggestions.
Kind regards,
Sophie Amiriantz

Reviewer 2 Report
Comments and Suggestions for Authors
1- Please modify the keywords to not be a repetition of the title.
2- Add the full name of any abbreviations in first mention as MBC
Author Response
Dear Reviewer,
First of all, I would like to thank you for reviewing our manuscript, as well as for your suggestions.
Here is how they have been addressed in the updated version of the manuscript:
- Please modify the keywords to not be a repetition of the title.
The keywords have been modified as follows:
Staphylococcus aureus; neomycin sulfate; ocular surface infection; antimicrobial resistance; treatment
2. Add the full name of any abbreviations in first mention as MBC
The full name of all abbreviations has been added prior to first mention (MBC and MIC).
Kind regards,
Sophie Amiriantz
Reviewer 3 Report
Comments and Suggestions for Authors
The manuscript from Amiriantz et al. finds that neomycin sulfate shows better anti-Staphylococcus aureus activity in EDTA, Tris and Polysorbate 80 solution. The author optimized that the composition of solution, found that 0.1% EDTA, 0.02% Tris and 0.1% Poly- 22 sorbate 80 could improve the efficacy of antimicrobial treatment in veterinary ocular surface infections. Using the new solution, they provided a new strategy to address the growing antimicrobial resistance issue worldwide. Even though the result looks interesting, the current study is lack of in-depth insights. It’s essential to perform more experiments to explain the current results for current manuscript. The current manuscript looks like preliminary results rather than scientific report. My questions/concerns are listed below.
1. The first of foremost of problem of current study is lack of rational design. They choose to use EDTA, Tris and Polysorbate 80, these are most commonly used reagents/buffers in biochemical lab. Why does the authors decide to use these reagents? And there are multiple substitutes, like MGDA, HEPES. What’s the effect of the analogues? They should compare side by side to figure out what’s the best combination for antimicrobial activity.
2. Another key concern is about the mechanism of current finding. The reagents in the solution do not have high antimicrobial potency, they work together to increase the activity of neomycin sulfate. What’s the mechanism of current finding. Does the new solution increase the solubility of neomycin sulfate? Or the solution helps neomycin sulfate penetrate the membrane of Staphylococcus aureus. The author should propose a mechanism and design proper experiments to prove that.
3. The format of abstract is not presented in professional manner. The authors should not list background, methods, results and conclusion in number.
4. There are some typos in the current manuscript, such as 50mM in line 171 should be 50 mM, and the concentration unit should be IU/mL in line 351, 359 and 371.
5. The current manuscript is lack of figures to illustrate their experiment design or key findings.
Comments on the Quality of English LanguageEnglish of current manuscript is fine, just minor editing of english language required.
Author Response

(The authors gave the same response as above.)

Reviewer 4 Report
Comments and Suggestions for Authors
In this study the authors presented their findings clearly indicating that the test solution did not exhibit significant bactericidal activity against the tested bacterial strains on its own. The authors disclosed a very interesting insight ie the potential synergy between the test solution and neomycin sulfate, highlighted the importance of combination therapies in treating bacterial infections.
However, the following points need to be addressed.
1. The manuscript lacks visual representations of the data, such as tables or graphs, to aid in understanding the results more easily for the readers.
2. The study acknowledges limitations, such as using standardized bacterial strains rather than clinical isolates and the need for further research on biofilm-forming bacteria.
3. Including potential clinical implications of the findings could enhance the relevance of the study, such as discussing how these results might inform future treatment strategies in veterinary ophthalmology.
Overall, the authors tried to bring in a comprehensive overview of the study methodology, results, and implications. Enhancing clarity, providing visual aids for data presentation, and discussing potential clinical implications could further strengthen the report and the manuscript is publishable in Antibiotics after addressing the above mentioned points.
Author Response
Dear Reviewer,
First of all, we would like to thank you for kindly reviewing our manuscript, as well as for the suggestions and comments you made about it.
You will find below our response to your comments.
1. The manuscript lacks visual representations of the data, such as tables or graphs, to aid in understanding the results more easily for the readers.
Understood, a graph (Figure 1) has been added to illustrate the impact of the test solution on the activity of neomycin sulfate against Staphylococcus aureus.
2. The study acknowledges limitations, such as using standardized bacterial strains rather than clinical isolates and the need for further research on biofilm-forming bacteria.
Yes, this is an improvement we would like to make to the design of upcoming studies.
3. Including potential clinical implications of the findings could enhance the relevance of the study, such as discussing how these results might inform future treatment strategies in veterinary ophthalmology.
Understood, the last paragraph of the discussion has been updated as follows:
In the current context of the emergence and spreading of antimicrobial resistance against commonly and less commonly used antibiotic compounds, new strategies to replace or reinforce antibiotics in veterinary ophthalmology are required. Treatment adjuvants such as an ocular cleaning solution containing Tris, EDTA and Polysorbate 80 could prove a valuable asset in veterinary ophthalmology and mitigate the complication risk leading to vision loss or enucleation. Some bacterial species (Pseudomonas aeruginosa, β-hemolytic Streptococcus spp.) have been identified as a risk factor for keratomalacia, a phenomenon leading to rapid worsening of corneal ulcers and possible destruction of the ocular globe [17] which requires aggressive medical treatment, or surgery [18,19]. Cleansing the ocular surface with a solution containing 0.02% Tris, 0.1% EDTA and 0.1% Polysorbate 80 could increase the permeability of bacteria involved in ulcerative disease to antibiotics administered topically. Reducing the concentrations of antibiotics required on the ocular surface to achieve bacterial eradication could contribute to preventing corneal ulcers from evolving into melting ulcers, and thus limit vision loss or the need for enucleation in veterinary patients.
We are hoping these answers and updates will provide satisfactory improvement to the manuscript.
Kind regards,
Sophie Amiriantz
Round 2
Reviewer 1 Report
Comments and Suggestions for Authors
The authors' responses have adequately addressed my inquiries and suggestions. I would like to recommend that this manuscript be considered for publication in the journal.
Comments on the Quality of English LanguageI have noticed some errors in the content, such as the scientific name of the microbe not being italicized.
Author Response
Dear reviewer,
The authors appreciate your feedback on the manuscript submitted and the responses to the first round of comments and suggestions.
The names of the microbes in the list of references have been italicized.
Kind regards,
Sophie Amiriantz
Reviewer 3 Report
Comments and Suggestions for Authors
The revised manuscript from Sophie Amiriantz improved the format and references of first draft. But my key concerns are not solved thoroughly. The revised manuscript is still short of proper control and mechanism study. The data in current report is so little that the conclusion they made may be not solid. I do not think the current manuscript suits for publishing in Antibiotics. My key concerns are listed here:
1. If the authors want to claim that EDTA, Tris and Polysorbate 80 are best combination for antimicrobial activity, they need design other common reagents as negative control.
2. For the mechnism study, I appreciate the authors list related references. I agree that with mechanism of EDTA antimicrobial activity. But this does not explain why do authors need combination and what is the effect of single component in antimicrobial activity?
3. For the figure they added, what is the Y-axis?
In all, the experiments in current manuscript are too superficial to make any scientific conclusions. They should submit the manuscript to other journals, rather than Antibiotics.
Author Response
Dear Reviewer,
The authors would like to thank you for your comments and suggestions regarding this manuscripts describing the studies. The preliminary nature of these studies has been better described in the updated manuscript, and its shortcomings have been highlighted in the title, abstract, discussion and conclusion. The authors understand that the findings of these studies cannot lead to as strong a conclusion as was initially proposed, and that additional, controlled studies with a larger sample size are needed to verify the present results. The authors will build on the reviewers’ comments and suggestions to design more appropriate studies in the near future for a better characterization and understanding of the test solution’s benefits.
- If the authors want to claim that EDTA, Tris and Polysorbate 80 are best combination for antimicrobial activity, they need design other common reagents as negative control.
Understood. The lack of a negative control has been commented in the discussion (lines 230-245).
- For the mechnism study, I appreciate the authors list related references. I agree that with mechanism of EDTA antimicrobial activity. But this does not explain why do authors need combination and what is the effect of single component in antimicrobial activity?
The authors understand this comment, and agree that a study evaluating the impact of each individual component is required to assess whether the combination tested is optimal.
- For the figure they added, what is the Y-axis?
The title of the Y-axis has been added.
In all, the experiments in current manuscript are too superficial to make any scientific conclusions. They should submit the manuscript to other journals, rather than Antibiotics.
The authors hope the changes made to the current manuscript make it fit for publication in Antibiotics, and will welcome any additional comment or suggestion.
Kind regards,
Sophie Amiriantz